# CTH/H_2_S Regulates LPS-Induced Inflammation through IL-8 Signaling in MAC-T Cells

**DOI:** 10.3390/ijms231911822

**Published:** 2022-10-05

**Authors:** Ting Lin, Xu Bai, Yuan Gao, Bohao Zhang, Jun Shi, Bao Yuan, Wenli Chen, Jianfu Li, Yong Zhang, Quanwei Zhang, Xingxu Zhao

**Affiliations:** 1College of Life Science and Biotechnology, Gansu Agricultural University, Lanzhou 730070, China; 2Gansu Key Laboratory of Animal Generational Physiology and Reproductive Regulation, Lanzhou 730070, China; 3College of Veterinary Medicine, Gansu Agricultural University, Lanzhou 730070, China

**Keywords:** clinical mastitis, H_2_S, CTH, inflammation, IL-8 signaling

## Abstract

Hydrogen sulfide (H_2_S), as an endogenous gaseous signaling molecule, plays an important role in the inflammatory process. Our previous study found that Cystathionine-γ-lyase (CTH) and H_2_S are correlated with the occurrence and development of Clinical Mastitis (CM) in Holstein cows. However, the functions and regulatory mechanisms of CTH/H_2_S are still unknown. In this study, the inflammatory mammary cell model based on the MAC-T cell line was established by Lipopolysaccharide (LPS)-induced manner to further explore the function and regulatory mechanism of CTH/H_2_S in cows with CM. In the inflammatory MAC-T cell, the CTH expression and H_2_S production were both repressed in an LPS-dose dependent manner, which demonstrated that CTH/H_2_S is related to the progression of inflammation. The inhibition of CTH/H_2_S using a selective CTH inhibitor, β-cyano-l-Alanine (BCA), promoted LPS-induced inflammation response and the expression of inflammatory cytokines. However, this was reversed by the H_2_S donor NaHS, demonstrating that H_2_S can protect cells from inflammatory damage. Intriguingly, interleukin-8 (IL-8) showed an inverse expression pattern correlated with the H_2_S-mediated cell protection effect during the inflammation process, and the inhibition test using a selective IL-8 receptor antagonist, SB225002, showed that IL-8 signaling plays a critical role in mediating endogenous H_2_S synthesis, and CTH/H_2_S exerts its anti-inflammation via IL-8-mediated signaling. This study provided support for the prevention and treatment of CM and the development of a novel anti-inflammatory strategy.

## 1. Introduction

Clinical mastitis (CM) is one of the greatest diseases impacting high milk yield and results in huge economic losses for the global dairy industry [1,2]. CM, a widespread inflammatory condition of the mammary glands, could be caused by a variety of risk factors, including host, pathogen, and environmental factors in dairy cattle. CM is easily monitored by veterinary clinical diagnosis (udder examination criteria: redness, hotness, swelling, and a painful sensation) in dairy cows [2]. However, it is still a great challenge to prevent and control CM for its complex etiology and multifactorial properties. Due to the capacity to eliminate a microorganism and the anti-inflammatory and regulatory effects on the microcirculation [3], antimicrobial or antibiotic agents use remains the main treatment strategy for CM in mammals. Reports showed that over half of antibiotics in the dairy industry are used for mastitis treatment and prevention [4]. Not surprisingly, antibiotic use is accompanied by a great concern of drug residues in milk, and the increased risks of allergies and drug-resistant pathogens, which raises the risk to human health and food safety. Thus, finding an alternative antimicrobial strategy and development of a novel prevention and treatment method is becoming urgent. Some novel treatment strategies including the development of vaccine, searching for alternative agents to antibiotics, and the exploration of traditional Chinese medicine are currently being developed [5]. However, it is still necessary to explore further advances in CM treatment and uncover the anti-inflammatory effects related to the various therapeutic strategies.

Hydrogen sulfide (H_2_S), as an endogenous gaseous signaling molecule, plays an important role in animal physiological regulation and exerts anti-bacterial activity [6,7]. Disturbed concentrations of H_2_S can lead to a variety of diseases related to the nervous system, cardiovascular system, digestive system and urinary system [8]. Studies revealed that H_2_S is involved in many physiological and pathological processes, including apoptosis, inflammatory and adaptive immunity such as monocyte and polymorphonuclear cell apoptosis, leukocyte adhesion and infiltration, T cell activation, proliferation, and inflammatory cytokine production [8,9]. H_2_S exerts its anti-inflammatory effects mainly via increasing the levels of anti-inflammatory and cytoprotective molecules and inhibiting the secretion of inflammatory cytokines [10]. Intriguingly, H_2_S may serve as a dual role modulator in the inflammatory process. Some reports showed that H_2_S can inhibit aortic atherosclerotic plaque and neutrophil aggregation in lung tissue, so as to reduce pulmonary edema and increase pulmonary capillary permeability [11]. H_2_S stimulates leukocyte activation and trafficking to promote a pulmonary inflammatory response in the model of septic shock, caused by cecal ligation and perforation [12]. These evidences indicate that H_2_S may serve as a dual role modulator in the inflammation process.

Cystathionine-γ-lyase (CSE/CTH) is a major endogenous H_2_S-producing enzyme with L-cysteine as the main substrate in mammalian cells. Either knockout or cell-specific overexpression of the *CTH* gene can evaluate the ability of H_2_S to modulate cellular functions [13]. Together with CTH, there are other enzymes involved in H_2_S production in mammals, namely, cystathionine-β-synthase (CBS), 3-mercaptopyruvate sulfotransferase (3-MST), and cysteine aminotransferase (CAT) [8,14]. These enzymes are widely distributed in mammalian tissues, including the liver, kidney, retina and spleen, and are induced by external stimulation and the intracellular environment [15]. The CTH/H_2_S system plays a part in both health and diseases, and CTH-derived H_2_S is implicated in the regulation of multiple signal physiological effects, including immune response, cell differentiation, cancer, aging and the injury process [6,16]. In addition, Cys, a non-essential amino acid with a nonpolar sulfhydryl group (-SH), is abundant in milk, and provides many substrates for endogenous H_2_S synthesis [17]. However, the underling mechanism of CTH/H_2_S-mediated inflammation regulation in mammary glands with CM is still unclear.

Our previous study identified a series of genes related to sulfide metabolism, including CTH and CBS in mammary glands with CM, and uncovered that CTH and H_2_S are correlated with the occurrence and development of CM in Holstein cows [18]. However, knowledge about the function and regulatory mechanism of CTH/H_2_S in Holstein cows is still unknown. This study aimed to find the expression pattern of CTH and the production property of H_2_S in the Lipopolysaccharide (LPS)-induced inflammation cell model, and uncover the molecular regulatory mechanism of CTH/H_2_S in the inflammation process, attempting to provide support for the prevention and treatment of CM and the development of an anti-inflammatory strategy.

## 2. Results

### 2.1. LPS Induction Inflammation Model in MAC-T Cells

As an endotoxin, LPS can effectively induce inflammation of bovine mammary glands [19]. To study the potential roles of H_2_S in mastitis in cattle, the bovine mammary alveolar cell inflammatory model was established using LPS in MAC-T cells. Expressions of multiple inflammatory factors, including *Interleukin-1β* (*IL-1β)*, *Interleukin (IL-8)*, *Interleukin-6 (IL-6)*, *Tumor Necrosis Factor-α* (*TNF-α)*, and *Toll Like Receptor 4 (TLR4)* were evaluated in both mRNA and protein levels after treatment, with scalar concentrations of LPS using a real-time quantitative polymerase chain reaction (qRT-PCR) and Western blot (WB) assay, respectively. Cell viability was evaluated by a Cell Counting Kit-8(CCK-8) assay. Normal MAC-T cells were set as controls. As shown in Figure 1, with the increase of LPS concentration, the MAC-T cell viability decreased gradually in an LPS dose-dependent manner (Figure 1A). Similarly, the transcripts of inflammatory factors *IL-1β* and *IL-8* increased gradually, in an LPS dose-dependent manner compared to the controls. In addition, the transcripts of *IL-6*, *TLR4* and *TNF-α* increased with the increase of the LPS concentration compared to the control group (Figure 1B–F). Consistent with the change of mRNA level, the protein expression of IL-8 and IL-1β also showed a gradual increase in an LPS dose-dependent manner. (Figure 1G–I). The results demonstrated that LPS can effectively induce an inflammatory response in MAC-T cells. IL-1β and IL-8 were determined as inflammatory markers in follow-up experiments.

### 2.2. CTH/H_2_S Are Repressed during LPS-Induced Inflammation Process

H_2_S as an important gaseous signaling molecule plays essential roles in the regulation of the inflammatory process [18,20]. To understand the production pattern of H_2_S and its potential function in mastitis, endogenous H_2_S production in supernatant was monitored by a Micro H_2_S Content Assay Kit, and the expression of key enzymes in H_2_S synthesis, including CTH and CBS, were evaluated by Immunofluorescence (IF), qRT-PCR and WB assays, respectively, in LPS-induced MAC-T cells. Normal MAC-T cells were set as controls. As shown in Figure 2, in normal MAC-T cells, the CBS with red fluorescence and CTH with orange fluorescence are distributed in the cytoplasm of all cells, in co-localization with CK18 marked by green fluorescence, which is the marker of the MAC-T cells (Figure 2A). Together with the fact that CTH and CBS are key enzymes in H_2_S synthesis, the result suggested that CTH and CBS are expressed in MAC-T cells, and H_2_S can be produced in the bovine mammary alveolar cell.

The detection of endogenous H_2_S concentration showed that a MAC-T cell can produce H_2_S, with an average concentration of 822 μM in supernatant (Figure 2B). However, the H_2_S production significantly decreased in LPS-induced inflammatory cells compared to normal MAC-T cells, even if the concentration of LPS is as low as 0.01 μg/mL (Figure 2B). Furthermore, with the increasing concentration of LPS treatment, the expression of *CTH* both at mRNA and protein levels showed a gradual decrease in an LPS dose-dependent manner (Figure 2C,E,F), which is consistent with the change of H_2_S production (Figure 2B). In addition, the transcripts of *CBS* were down-regulated in the LPS-induced inflammatory cells compared to the control group (Figure 2D). However, CBS expression showed different changes compared to normal MAC-T cells, (Figure 2E,G). These results indicated that CTH and CBS expression were repressed in inflammation, which lead to the inhibition of H_2_S synthesis. Notably, CTH/H_2_S is closely related to the progression of inflammation.

LPS with a concentration of more than 100 μg/mL would greatly impact cell viability and H_2_S production in MAC-T cells. Therefore, an LPS concentration of 100 μg/mL was determined as inflammation-induced concentrations in follow-up experiments.

### 2.3. CTH/H_2_S Rescued Cell Survival during Inflammation Process

To further explore the roles of CTH/H_2_S in the inflammation process, β-cyano-l-Alanine (BCA), a CTH-specific reversible inhibitor [21,22], and NaHS, a H_2_S donor to trigger H_2_S release in cells [23], were introduced to uncover the CTH/H_2_S function. *CTH* and *CBS* expression was evaluated by qRT-PCR and WB assays. Endogenous H_2_S production in supernatant was determined by a Micro H_2_S Content Assay Kit, and cell viability was evaluated by CCK-8 assay. Normal MAC-T cells without treatment were set as controls. As shown in Figure 3, LPS treatment inhibited *CTH* expression both at the mRNA level and protein levels in MAC-T (Figure 3A,C,D).

BCA treatment with a 1 mM (Appendix A) further significantly downregulated *CTH* expression both at mRNA and protein levels compared to the control and LPS treatment group (Figure 3A,C,D). In contrast, the addition of NaHS can rescue the inhibition of CTH induced by LPS and BCA (Figure 3A,C,D). Additionally, the expression of CBS is similar to that of CTH. *CBS* was also repressed by LPS and BCA, but rescued by NaHS both at mRNA and protein levels (Figure 3B,C,E). The results indicated that BCA may lead to down-regulation of the expression of CTH and CBS in H_2_S synthesis, but the H_2_S donor could rescue H_2_S production via CTH and CBS. Consistently, the monitoring of H_2_S synthesis and cell survival rate showed that LPS and BCA treatment suppressed endogenous H_2_S production and cell viability in MAC-T, and the decrease was rescued by NaHS (Figure 3F,G). These results demonstrated that inhibition of H_2_S production aggravates the decline in cell viability during the inflammation process, and the introduction of H_2_S using NaHS can protect cells from inflammatory damage.

### 2.4. H_2_S Repressed Inflammatory Cytokines Expression

To further find the regulation effect of CTH/H_2_S on the expression of inflammatory cytokines, *IL-1β* and *IL-8* expression were monitored by qRT-PCR and WB assays after LPS, BCA or/and NaHS treatment. Normal MAC-T cells without treatment were set as controls. As shown in Figure 4, IL-1β and IL-8 expression were significantly stimulated in the LPS-induced inflammation process in MAC-T. Intriguingly, the block of endogenous H_2_S synthesis with BCA treatment promoted IL-8 expression, and the addition of H_2_S donor, NaHS, inhibited *IL-8* expression both at mRNA and protein levels (Figure 4A,C,D), indicating that H_2_S production repressed IL-8 expression, consistent with the previous result that H_2_S production can protect cells from inflammatory damage. However, IL-1β expression was reduced after BCA treatment, and NaHS did not reverse this situation, which is inconsistent with the previous data. An explanation could be that the anti-inflammatory effect of the CTH/H_2_S system does not depend on IL-1β. It is possible that IL-1β has a direct inhibitory effect, and its specific mechanism of action remains to be further explored (Figure 4B,C,E). These results showed that the IL-8 expression pattern was inversely correlated with the H_2_S-mediated cell protection effect during the inflammation process, raising the hypothesis that H_2_S may act on its anti-inflammation mainly via the IL-8-mediated manner.

### 2.5. CTH/H_2_S Acts Anti-Inflammation Effect via IL-8 Signaling

IL-8 is one of the major mediators of the inflammatory response [24]. To further confirm the role of IL-8 in LPS-induced inflammation and H_2_S-induced anti-inflammation effects, SB225002, a selective IL-8 receptor antagonist [25], was introduced to blocking IL-8-mediated signaling. After treatment, cells were collected for a cell viability test, H_2_S concentration monitor, CTH/CBS and IL-8/IL-1β expression detection, respectively. Normal MAC-T cells without treatment were set as controls. As shown in Figure 5, different concentrations of SB225002 were added to the LPS-induced inflammatory MAC-T cells. The result showed that SB225002 with a concentration of 0.5 μM or less can effectively reverse the LPS-induced reduction of cell viability. When the concentration of SB225002 exceeds 0.5 μM, the cell survival rate decreases significantly, due to the negative effects caused by a high concentration of SB225002 (Figure 5A). Therefore, the SB225002 concentration of 0.5 μM was determined as inflammation-induced concentrations in follow-up experiments. The monitor of endogenous H_2_S production showed that blocking IL-8 signaling using its receptor antagonist SB225002 significantly increased H_2_S synthesis compared to the LPS-induced inflammation group (Figure 5B). In addition, the SB225002 treatment and NaHS supplement both promote *CTH* and *CBS* expression in mRNA and protein levels compared to the LPS-induced inflammation group (Figure 5C–G), consistent with the endogenous H_2_S production pattern. These results demonstrated that IL-8 signaling plays a critical role in mediating endogenous H_2_S synthesis.

Further mechanism studies showed that the SB225002 treatment decreased *IL-8* expression in mRNA and protein levels, compared to both the control and LPS-induced inflammation group (Figure 6A,C,D). Moreover, the addition of the H_2_S donor also obviously inhibited IL-8 expression (Figure 6A,C,D). Whereas, although the LPS-stimulated IL-1β expression can also be repressed by the SB225002 and H_2_S donor, IL-1β expression is not lower than the control (Figure 6B,C,E). These results support the hypothesis that CTH/H_2_S exerts its anti-inflammation via IL-8-mediated signaling. The inhibition of the CTH/H_2_S system can lead to increased IL-8 expression and aggravate the inflammatory response. Conversely, after inhibiting IL-8 expression, the anti-inflammatory effect of CTH/H_2_S can be promoted (Figure 6F).

## 3. Discussion

CM is considered the most common disease in dairy farming, causing economic losses to the dairy industry due to reduced milk production and poor quality. Currently, the treatment of CM infections mainly relies on antibiotics. However, the widespread use of antibiotics has led to an increased risk of the emergence of allergic and drug-resistant pathogens, and poses ongoing risks to human health and food safety. Therefore, the need to develop alternative drugs and treatments for the dairy industry is urgent [26]. H_2_S, as a gaseous transmitter, is mainly synthesized by CTH, which has broad substrate specificity and can catalyze cysteine to pyruvate, ammonia and thiocysteine. Thiocysteine can be further catalyzed by CTH to produce H_2_S [13]. In addition, CBS, 3-MST and CAT are also involved in producing H_2_S [27]. The CTH/H_2_S system plays a part in both health and diseases, and is implicated in the regulation of multiple physiological and pathological effects, including inflammation, immune response, cell differentiation, cancer, aging and the injury process [6,13,16].

Our previous study identified a series of differentially expressed proteins associated with H_2_S metabolism from Holstein cows with CM, including CTH and CBS. The study confirmed that endogenous H_2_S production in the serum, and the expression of CTH and CBS, are closely correlated with the occurrence and development of CM in cows [18]. In this study, we established an inflammatory mammary cell model using MAC-T cell line by an LPS-induced manner (Figure 1), to further explore the function and regulatory mechanism of CTH/H_2_S in cows with CM.

As well-known inflammatory cytokines or chemokines, IL-1β, IL-8, IL-6, TNF-α and TLR4 are involved in various types of inflammatory responses, and play key roles in the inflammatory process during bacterial infection of bovine mammary glands. The results showed that the expression of the above-mentioned inflammatory factors significantly increased after LPS treatment (Figure 1), which indicated that the LPS-induced MAC-T cell inflammation model was successful. In the LPS-induced inflammatory MAC-T cell, the CTH expression and H_2_S production were both repressed in an LPS dose-dependent manner (Figure 2). This demonstrated that CTH/H_2_S is related to the progression of inflammation. In addition, the inhibition of CTH/H_2_S using a selective CTH inhibitor, BCA, promoted the LPS-induced inflammation response and the expression of inflammatory cytokines. In contrast, the H_2_S donor, NaHS, suppressed the LPS-induced inflammation response and the expression of inflammatory cytokines (Figure 3 and Figure 4), demonstrating that H_2_S can protect cells from inflammatory damage. Our study is consistent with previous reports that H_2_S showed an anti-inflammatory effect in various tissues and organs in mammals, such as acute lung inflammation [28], vascular inflammation [29], cardiovascular inflammation [30,31], and airway inflammation [32].

Intriguingly, the inflammatory cytokine, IL-8, showed an inverse expression pattern correlated with the H_2_S-mediated cell protection effect during the inflammation process (Figure 4), raising the hypothesis that H_2_S may act its anti-inflammation mainly via the IL-8-mediated manner. The results of the IL-8 signaling inhibition test showed that IL-8 signaling plays a critical role in mediating endogenous H_2_S synthesis, and CTH/H_2_S exerts its anti-inflammation via IL-8-mediated signaling (Figure 5 and Figure 6). Nevertheless, there are still some problems that need to be solved, for example, if H_2_S is competitively binding the IL-8 receptor, and how IL-8 plays in H_2_S-mediated anti-inflammatory effects. The underlying mechanism is still unclear and need further investigation. In addition, the mechanism of H_2_S-mediated anti-inflammatory effects has been very limited. Previous reports indicated that some proteins and signals were involved in the H_2_S-mediated anti-inflammatory process. H_2_S inhibits aortic valve calcification and inflammation in cardiovascular via Nuclear factor kappa-B (NF-κB) establishing a link between inflammation and mineralization in vascular calcification [30]. H_2_S stimulates autophagy by blocking Mammalian target of rapamycin (mTOR) signaling in sepsis-induced acute lung injury [33]. H_2_S promotes activation of Nuclear factor-E2 related factor2 (Nrf2) and the consequent expression of antioxidant proteins to alleviates liver damage [34]. Endogenous H_2_S can suppress the P2X7 receptor (P2X7R) to attenuate NOD-like receptor thermal protein domain associated protein 3 (NLRP3) inflammasome-mediated neuroinflammation in rats [35]. These findings illustrate that H_2_S can inhibit inflammation through multiple pathways, but the molecular targets of its action and its mixed effects are not fully understood. In addition, H_2_S has a regulatory effect on the release and function of various inflammatory factors. Therefore, regardless of whether H_2_S has pro-inflammatory or anti-inflammatory effects, the development of H_2_S systemic administration using inhibitors of H_2_S synthesis or through H_2_S donors can provide new avenues for the treatment of inflammatory diseases. This study revealed a novel downstream target of hydrogen sulfide in the IL-8 signaling pathway. All these findings help to understand how hydrogen sulfide works, and provides a basis for hydrogen sulfide as a drug development and application.

## 4. Materials and Methods

### 4.1. Cell Culture and Treatment

MAC-T cells were purchased from the ATCC Corporation (Beijing, China), and maintained in dishes or plates in a humidified atmosphere of 5% CO_2_ at 37 °C in a DMEM culture medium (Gibco, Grand Island, NY, USA) containing 10% fetal bovine serum (FBS, Gibco, Grand Island, NY, USA). The cells were grown to 80% confluence, and cells were passaged or treated with different compounds such as LPS (0–200 μg/mL, Solarbio, Beijing, China), β-cyano-L-Alanine (BCA, 0.5–4.0 mM, Cay-man Chemical, Ann Arbor, MI, USA), NaHS (30 µM, Sigma, St. Louis, MO, USA), or/and SB225002 (0–10 μM, Selleck, Houston, TX, USA). The cells were seeded at 1 × 10^4^ cells/well in 96-well plates or 1 × 10^5^ cells/cm^3^ in different dishes. Before treatment, the culture medium was changed to DMEM without FBS and phenol (Gibco) for 12 h. The MAC-T cells were treated with or without LPS, BCA, NaHS, or/and SB225002 for 24 h. The appropriate concentration gradient was selected based on literature reports [36,37,38] and cell viability test results, and adjusted according to the reagent situation. Each treatment was replicated at least in triplicate.

### 4.2. Endogenous H_2_S Detection

The endogenous H_2_S concentration of cells culture supernatants were tested following the manufacturer’s instructions of the Micro H_2_S Content Assay Kit (Solarbio, Beijing, China). All H_2_S detection was evaluated under a wavelength of 665 nm using a microplate reader (ReadMax 1900, Shanghai, China). All experiments were conducted in triplicate and repeated at least three times.

### 4.3. Cell Viability Test

The viability of MAC-T cells was measured after treatment with or without LPS (0–200 μg/mL), BCA (0.5–4.0 mM), NaHS(30 μM) and SB225002 (0–10 µM) using a CCK-8, (Bimake, Shanghai, China) according to the manufacturer’s instructions. Briefly, the cells were seeded in 96-well plates and incubated for 24 h. The cells were treated with different concentrations of LPS, BCA, NaHS and SB225002 for 24 h. 10 µL of CCK-8 was added to the cells and allowed to incubate for 2 h before measuring the OD at 450 nm with a ReadMax 1900 microplate reader (Flash, Shanghai, China). All concentration gradient selections are based on the literature search and adjusted according to reagent conditions. The cell viability experiments were performed at least in triplicate.

### 4.4. IF Assay

MAC-T cells were plated in a 35 mm dish with cover slips to a monolayer. IF assay was performed as previous described [39]. In brief, the slips with monolayer cell were fixed in 4% formaldehyde for 30 min and permeabilized in 0.1% Triton X-100 for 15 min at room temperature. After being blocked with 5% normal donkey serum (Solarbio, Beijing, China), cover slips were incubated with primary antibodies as follows: mouse monoclonal anti-CTH (1:150, Abcam, Cambridge, UK), rabbit anti-CBS (1:400, Abcam, Cambridge, UK), and mouse anti-cytokeratin 18 (CK18, 1:100, Bioss, Beijing, China) at 4 °C overnight. This was followed by incubation with an Cy5-labeled goat anti-rabbit IgG (1:350; Bioss, Beijing, China), a FICT-labeled rabbit anti-mouse (1:350; Bioss, Beijing, China) and a Cy3-labeled rabbit anti-mouse (1:350; Bioss, Beijing, China) for 1 h at 37 °C. The nuclei were localized using 4,6-diamidino-2-phenylindole (DAPI, Solarbio, Beijing, China). Images were captured with a digital camera under a Zeiss LSM800 confocal microscope (Carl Zeiss, Oberkohen, Germany).

### 4.5. RNA Isolation, cDNA Synthesis, and qRT-PCR Assays

The cellular total RNA was extracted using a FastPure RNA isolation kit (Vazyme, Nanjing, China) according to the manufacturer’s protocols. RNA quantify was monitored by a NanoDrop-8000 (ThermoFisher Scientific, Waltham, MA, USA), and RNA integrity was assessed by 1% denaturing gel electrophoresis (Biowest Regular Agarose, Castropol, Spain). The cDNA was synthesized using the Evo M-MLV RT Kit (Agbio, Hunan, China). The qRT-PCR was performed with 2×SYBR Green *pro Taq* HS Premix (Agbio, Hunan, China) following the manufacturer’s protocols. The amount of transcripts in each sample were normalized using *Glyceraldehyde-3-phosphate dehydrogenase* (*GAPDH*) as the internal control. Primers used for qRT-PCR assay (Appendix A) were designed by Premier software (version 5.0) and purchased from Qinke Biotech (Yangling, China). Thermo cycle was conducted on the Light Cycler 96 real-time system (Roche, Basilea, Switzerland). All qRT-PCR assays were performed at least in triplicate.

### 4.6. Western Blot Assay

After treatment, MAC-T cells were harvested for protein extraction using a Radio-immunoprecipitation assay buffer (RIPA, Solarbio, Beijing, China) with 1 mM phenylmethylsulfonyl fluoride (PMSF, Solarbio, Beijing, China) and a protease inhibitor (PI, Solarbio, Beijing, China). The protein concentration was determined following a Bicinchoninic Acid (BCA) protein assay kit (Boster, Wuhan, China). A WB assay was performed as previous described [40]. Briefly, equal total protein samples (30 μg) were loaded and separated via 12% sodium dodecyl sulfate-polyacrylamide gel (SDS-PAGE). The blots were electro-transferred onto a polyvinylidene fluoride (PVDF) membrane (Millipore, Boston, MA, USA) and blocked with 5% (*w*/*v*) of skimmed milk in a Tris-HCl buffer (Solarbio, Beijing, China) at room temperature. Co-incubation of blocked PVDF overnight at 4 °C with the following primary antibodies: rabbit anti-IL-1β (1:300, Bioss, Beijing, China), mouse monoclonal anti-IL-8 (1:200, Santa, Dallas, Texas USA), rabbit anti-CBS (1:3000, Abcam, Cambridge, UK), mouse monoclonal anti-CTH (1:1000, Abcam, Cambridge, UK), and anti-β-actin (1:4000, Bioss, Beijing, China). After three washes, the membranes were incubated with the corresponding secondary antibody conjugated to horseradish peroxidase (Bioss, Beijing, China) for 1 h at room temperature. Blot bands were visualized using the Gel Imaging System (Tannon Science & Technology, Shanghai, China) and then digitized using Image-Pro Plus 6.0 (Media Cybernetics, Rockville, MD, USA). β-actin was used as a control. All immunoblot assays were performed at least in triplicate.

### 4.7. Statistical Analysis

All statistical procedures were performed using Statistical Package for the Social Sciences 22.0 for Windows (SPSS Inc., Chicago, IL, USA). Graphs were constructed using Prism 5.0 (GraphPad Software Inc., San Diego, CA, USA). One-way ANOVA (Analysis of variance) and post hoc Tukey’ test were used for finding significant differences between groups. *p* < 0.05 were considered as significant.

## 5. Conclusions

In MAC-T cells, LPS can significantly reduce the content of H_2_S and significantly increase the expression of IL-8, IL-β, TLR4 and other inflammatory factors. The relative expression levels of *CTH* mRNA and protein and the content of H_2_S were inhibited in a LPS dose-dependent manner, and were negatively correlated with the relative expression levels of IL-1β and IL-8. Cells were treated with a CTH inhibitor BCA, and the results indicated that CTH inhibited the development of inflammation by regulating the LPS-induced reduction of H_2_S synthesis and the expression of IL-8 in MAC-T cells. Inhibition assays using the selective IL-8 receptor antagonist SB225002 suggest that IL-8 signaling plays a critical role in mediating endogenous H_2_S synthesis, and that CTH/H_2_S exerts its resistance through IL-8-mediated signaling. These results can provide new ideas and methods for the prevention and treatment of clinical mastitis and the development of anti-inflammatory drugs.

## Figures and Tables

**Figure 1 ijms-23-11822-f001:**
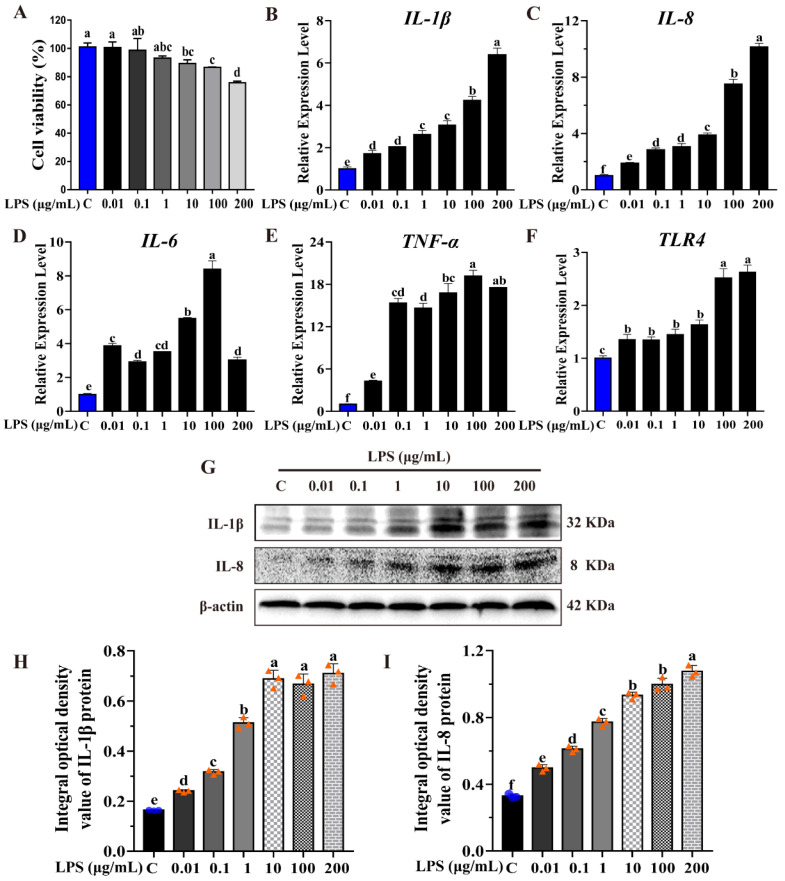
LPS induction established inflammation model in MAC-T cells. (**A**) MAC-T cell viability was monitored by CCK-8 assay after LPS treatment with different concentration. (**B**–**F**) Relative expression level of *IL-1β*, *IL-8*, *IL-6*, *TNF-α* and *TLR4* were evaluated by qRT-PCR after LPS treatment with different concentration, respectively. (**G**) IL-1β and IL-8 protein expression were detected by Western blot after LPS treatment with different concentration. (**H**,**I**) Blot bands of IL-1β and IL-8 were digitized for optical density (OD) value using Image-Pro Plus 6.0, respectively. The different lowercase letters above the bars indicate a significant difference in different treatment groups (*p* < 0.05).

**Figure 2 ijms-23-11822-f002:**
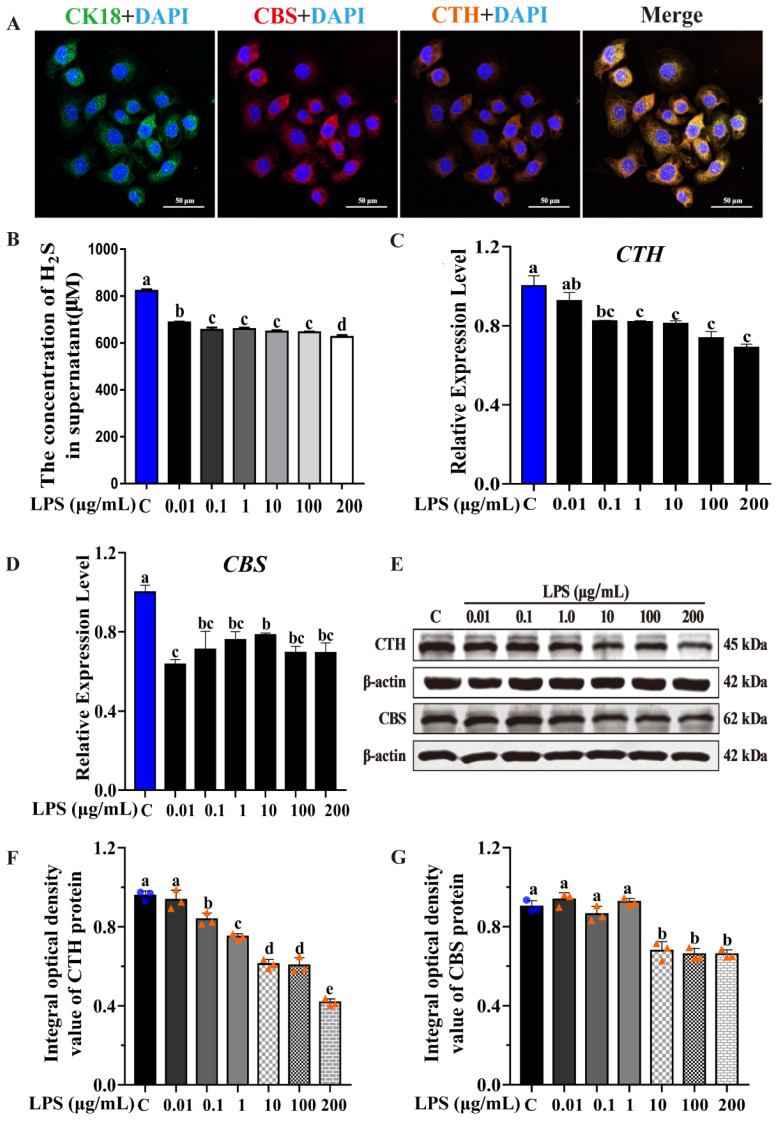
CTH expression and H_2_S production are repressed in LPS-induced inflammatory process. (**A**) Location analysis of CBS, CTH, and CK18 proteins in the MAC-T cells (400×). (**B**) H_2_S production in supernatant was monitored by Micro H_2_S Content Assay Kit after LPS treatment with different concentration. (**C**,**D**) Relative expression level of *CTH* and *CBS* were evaluated by qRT-PCR after LPS treatment with different concentration, respectively. (**E**) CTH and CBS protein expression were detected by Western blot after LPS treatment with different concentration. (**F**,**G**) Blot bands of CTH and CBS were digitized for OD value using Image-Pro Plus 6.0, respectively. The different lowercase letters above the bars indicate a significant difference in different treatment groups (*p* < 0.05).

**Figure 3 ijms-23-11822-f003:**
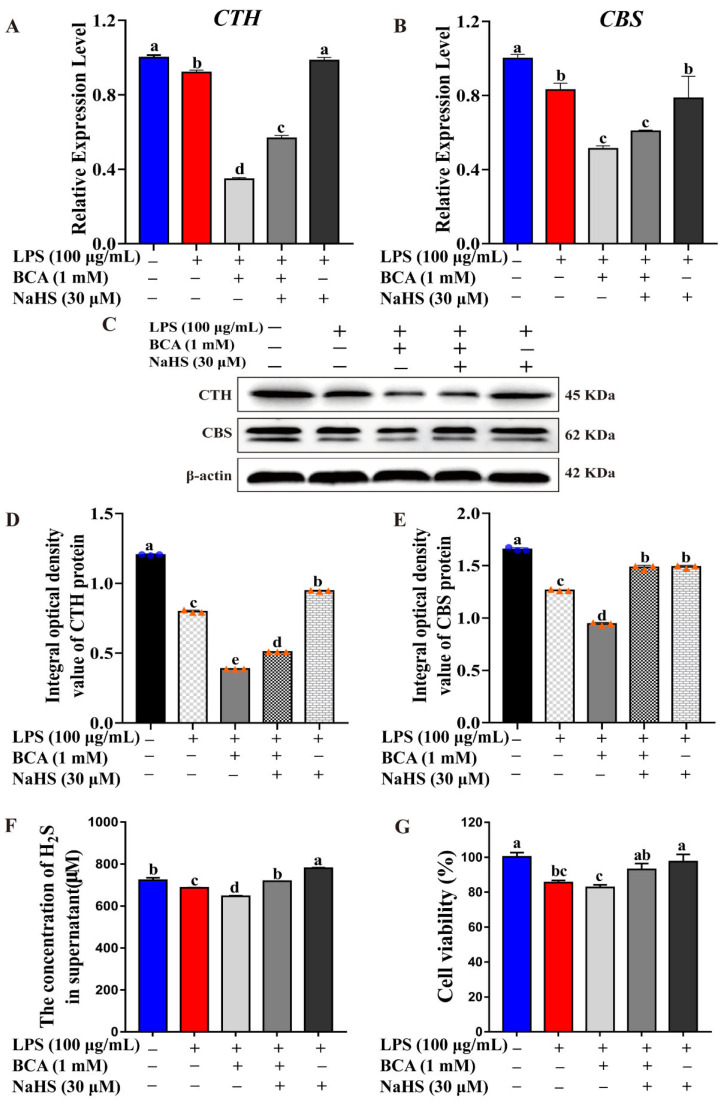
CTH/H_2_S rescued cell survival during inflammation process. (**A**,**B**) Relative expression level of *CTH* and *CBS* were evaluated by qRT-PCR after LPS, BCA and/or NaHS treatment, respectively. (**C**) CTH and CBS protein expression were detected by Western blot after LPS, BCA and/or NaHS treatment, respectively. (**D**,**E**) Blot bands of CTH and CBS were digitized for OD value using Image-Pro Plus 6.0, respectively. (**F**) H_2_S production in supernatant was monitored by Micro H_2_S Content Assay Kit after LPS, BCA and/or NaHS treatment, respectively. (**G**) Cell viability was monitored by CCK-8 assay after LPS, BCA and/or NaHS treatment, respectively. The different lowercase letters above the bars indicate a significant difference in different treatment groups (*p* < 0.05).

**Figure 4 ijms-23-11822-f004:**
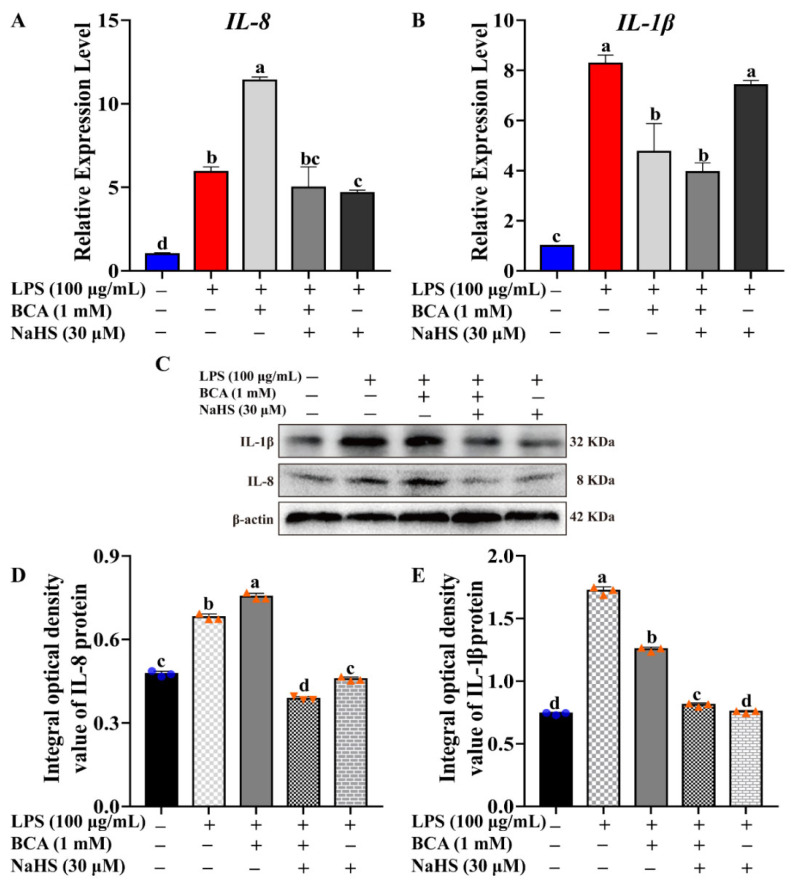
H_2_S repressed inflammatory cytokines expression. (**A**,**B**) Relative expression level of *IL-8* and *IL-1β* were evaluated by qRT-PCR after LPS, BCA and/or NaHS treatment, respectively. (**C**) IL-8 and IL-1β protein expression were detected by Western blot after LPS, BCA and/or NaHS treatment, respectively. (**D**,**E**) Blot bands of IL-8 and IL-1β were digitized for OD value using Image-Pro Plus 6.0, respectively. The different lowercase letters above the bars indicate a significant difference in different treatment groups (*p* < 0.05).

**Figure 5 ijms-23-11822-f005:**
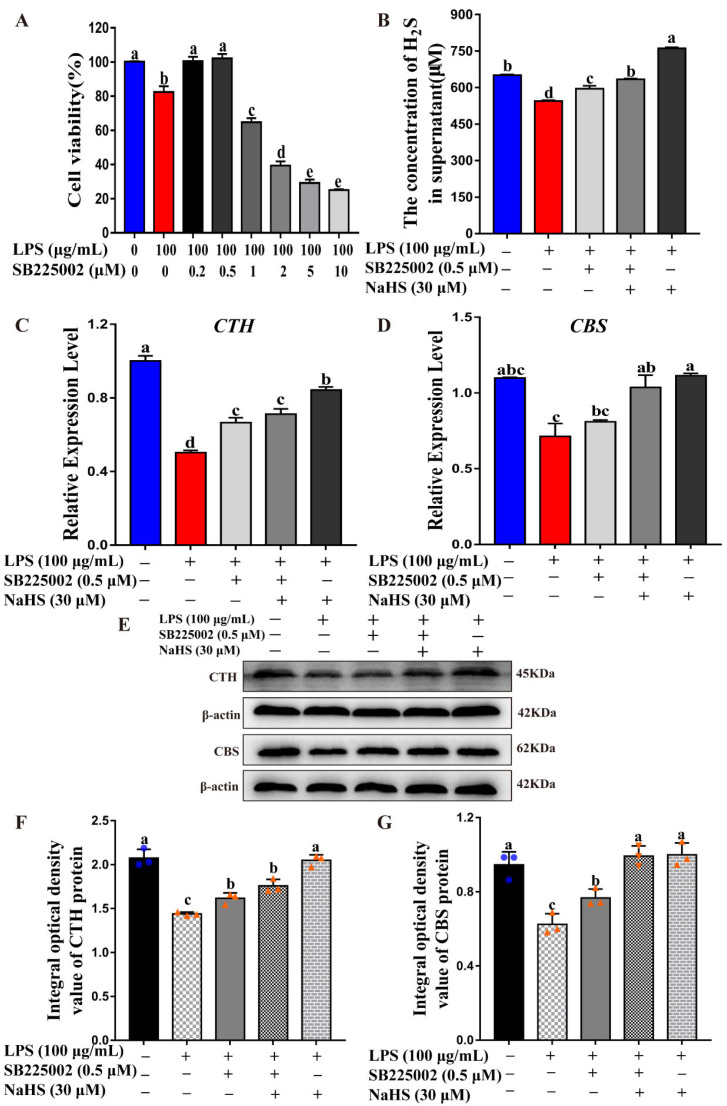
Blocking IL-8 signaling significantly increased H_2_S synthesis and CTH/CBS expression. (**A**) Cell viability after IL-8 receptor antagonist SB225002 treatment with different concentration was monitored by CCK-8 assay. (**B**) H_2_S production in supernatant was monitored by Micro H_2_S Content Assay Kit after LPS, SB225002 and/or NaHS treatment, respectively. (**C**,**D**) Relative expression levels of *CTH* and *CBS* were evaluated by qRT-PCR after LPS, SB225002 and/or NaHS treatment, respectively. (**E**) CTH and CBS protein expression were detected by Western blot after LPS, SB225002 and/or NaHS treatment, respectively. (**F**,**G**) Blot bands of CTH and CBS were digitized for OD value using Image-Pro Plus 6.0, respectively. The different lowercase letters above the bars indicate a significant difference in different treatment groups (*p* < 0.05).

**Figure 6 ijms-23-11822-f006:**
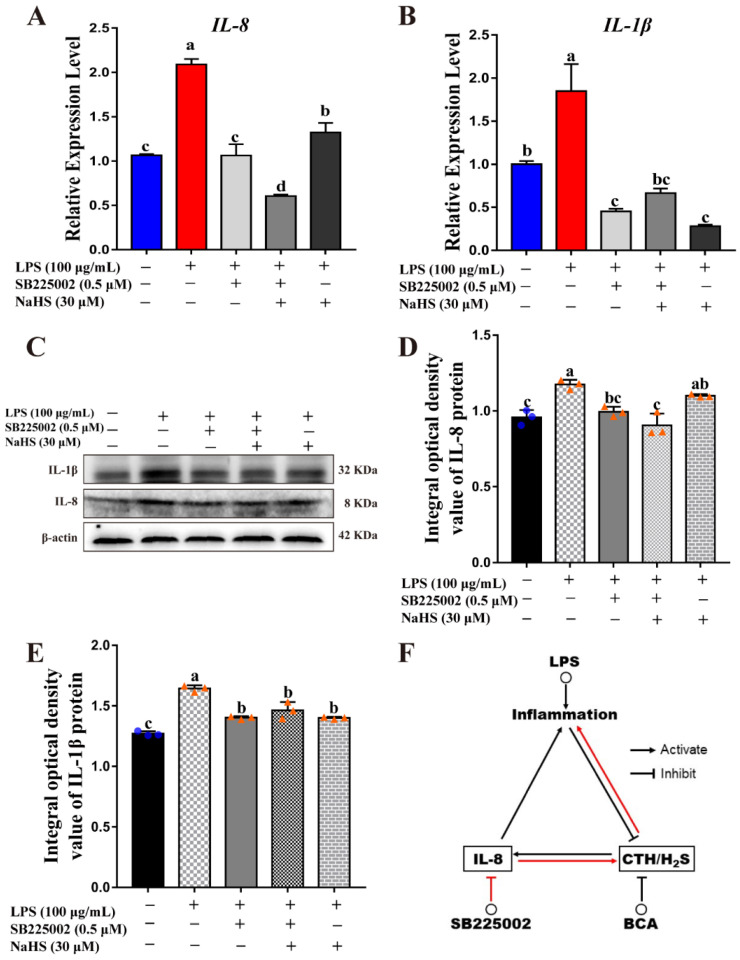
SB225002 treatment inhibited IL-8 and IL-1β expression. (**A**,**B**) Relative expression level of *IL-8* and *IL-1β* were evaluated by qRT-PCR after LPS, SB225002 and/or NaHS treatment, respectively. (**C**) IL-8 and IL-1β protein expression were detected by Western blot after LPS, SB225002 and/or NaHS treatment, respectively. (**D**,**E**) Blot bands of IL-8 and IL-1β were digitized for OD value using Image-Pro Plus 6.0, respectively. (**F**) The interaction of CTH/H_2_S and IL-8, different colored arrows represent different pathways. The different lowercase letters above the bars indicate a significant difference in different treatment groups (*p* < 0.05).

## Data Availability

The data that support the findings of this study are available from the corresponding author upon reasonable request.

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
