# Peer review of "CTH/H2S Regulates LPS-Induced Inflammation through IL-8 Signaling in MAC-T Cells"

_ijms, 2022, doi:10.3390/ijms231911822_

Round 1
Reviewer 1 Report
In the paper “CTH/H2S regulates LPS-induced inflammation through IL-8 signaling in MAC-T cells”
the authors aim to investigate the role played by H2S and related enzymes (CTH and CBS) in the LPS-induced inflammatory status in a bovine mammary gland cell line (MAC-T) at both genomic and proteomic levels.
The paper is of interest, nonetheless, some major issues arise:
- The English text should be improved in order to make it easier to read;
- The abstract should be improved (too long sentences)
- Some parts of the discussion need to be better explained (see below)
- In the Material and Methods section some more information should be added (see below)
- The conclusion should be improved by adding practical/therapeutic implications of the obtained results (currently, it appears to be a repletion of the discussion).
For these reasons, the paper can be accepted only after major revisions.
Introduction
Page 1 line 33: change “inflammatory” as “inflammatory condition”
Page 1 line 37: what do the authors mean by “by visible abnormalities in dairy cows”? Explain it better
Page 2 line 48: change “about” as “related to”
Page 2 lines 50-51: what do the authors mean by “exerts a universal defense against antibiotics in bacteria”?
Page 2 lines 58-59: what do the authors mean by “Intriguingly, H2S may serve as a dual-role modulator in inflammatory process”? Are the following sentences the explanation? It is not clear. Specify better
Page 2 lines 66-67: what do the authors mean by “CTH gene has enabled a unique appreciation of the ability of H2S to to modulate cellular functions”?
Results
General consideration: it is not clear if the authors comment the results related to the control cells culture or the one treated with LPS. This should be checked and better explained throughout the section.
Page 2 line 88: remove “established” from the title of the section
Page 2 line 91: remove “induced manner” after “LPS”
Page 3 line 93-94: change the sentence “after LPS treatment with gradient concentrations” as “after treatment with scalar concentrations of LPS”
Page 3 line 98: the authors state that all the tested cytokines show a gradual increase. This affirmation is not true for IL-6 where by increasing LPS concentrations a peak is present for 100 ug/mL followed by a consistent decrease with LPS 200 ug/mL. Could the authors explain this results?
Page 3 Figure legend: remove “by Student’s t-test (between two groups) or”
Page 4 line 115: change “played” as “plays”
Page 4 lines 120-122: the sentence “In normal MAC-T cells, the CBS, marked 120 by red and CTH, marked by orange, showed widely distributed in all cells, overlapping with CK18 labeled with green, which is the marker of MAC-T cells (Fig. 2 A)” is not clear. Rephrase it
Page 4 line 128: the sentence “gradually, in an LPS dose-dependent manner” is not correct as the H2S concentration significantly decrease after LPS 0.01 ug/mL, but with higher concentrations do not change.
Page 6 line 160: why di the authors state that the decrease in CBS is unexpected?
Page 6 lines 166-168: the sentence is not clear. Do the authors mean NaHS instead of H2S (line 168)?
Page 8 line 183: remove further
Page 8 line 186: remove H2S after production
Page 8 lines !87-191: these sentence and explanation are not clear. Please rephrase. Moreover, NaHs is not able to restore IL-1b gene expression if compared to LPS+BCA treated cells
Page 9: do the author could give an explanation about the increase of cytotoxicity exerted by SB225002 concentrations higher than 0.5 uM?
Page 10 line 227: delete “obviously”
Discussion
Page 11 lines 242-244: the sentence “Due to that antibiotic use is accompanied by a great concern of drug 243 residues, leading to the increased risks of allergies and drug-resistant pathogens” is not clear
Page 11 line 244-245: what do the authors mean with “A lack 244 of such drugs and the target molecules of CM”?
Page 12 lines 278-290: this part should better discussed.
Material and methods
In general, more information related to the treatment and co-treatment of cell line should be addressed. Is not clear when BCA, NaHS and SB225002 have been applied to cultures and how much time the cells were exposed. How many cells have been plated in each well in the different experiments? How many replicates? What is the rationale to choose the range of concentrations?
Reviewer 2 Report
This is an interesting paper, publishable in the journal in its present form, according to my criteria; provided some minor English re-style is taken care of; beware particularly of the use of the past tense
Reviewer 3 Report
This manuscript reports the results of in vitro experiments with mammary epithelial cells to examine the regulation of inflammation driven by LPS by hydrogen sulfide.
Both the introduction and discussion present clinical mastitis and antimicrobials and then moves to anti-inflammatory actions of hydrogen sulfide. There is a missing piece between clinical mastitis and the need for anti-inflammatory actions. It seems counterintuitive to reduce the response to infection if antimicrobial use is being limited. Wouldn’t it be better to stimulate inflammation to fight the infection? Strengthen your argument. Make the connection.
A major improvement to the manuscript is possible using a Tukey multiple comparison, or other similar, following ANOVA. T-tests for pairwise comparisons don’t protect for multiple comparisons as Tukey does. This would allow the use of superscripts to identify differences among treatments and eliminate the need for brackets that make figures busy. Application of Tukey and installation of superscripts will greatly improve this manuscript’s data presentation and interpretation.
There are some wording choices that should be examined and revised. See line 50-51 for bacteria antibiotics, line 76-77 for mediating inflammation (regulating?) and others. Accurately conveying your message should be improved.
Authors state triplicates and three wells. Clarify how many observations are present. How were treatments replicated and what was the experimental unit (well?)? Be more clear.
Figure 1G shows proteins not reported in the western blot methods. Add antibodies to methods description.
RNA measurements utilized only 1 housekeeping gene and are unclear on how it was selected. Were primers validated - single product? Are primers intron spanning? Was efficiency of the primers measured?
A summary figure/diagram for how authors believe the resulting interactions of proteins would be helpful. There appear to be some circular forms of regulation and a figure would help clarify if that’s what authors are saying such as inhibiting the IL-8 receptor inhibits IL-8 production and more hydrogen sulfide leads to more CTH.
Figure 2B shows LPS effects on hydrogen sulfide. Authors state a dose response (i.e. line 364 and elsewhere) but any level of LPS, even the minimum resulted in a similar reduction in hydrogen sulfide and this does not support a dose response. This response is characterized by presence vs. absence not an effect of increasing level. Clarify statements related to this.
Line 122-125 - this conclusion is not fully supported. It is circular. Revise the statement for clarity. Hydrogen sulfide presence doesn’t necessarily indicate high expression or that activity is high in general, not without a comparison to something else.
Line 126 says 750 but the figure appears to show 800.
Line 162 - how was this inference made? The regulation of one factor by the other was not directly evaluated by this experimental design. Revise this statement.
Line 166 mentions cell damage, but no measurements of cell damage are reported. For the results, adhere to the results that are specifically measured.
Line 187 - consider the interpretation of these results. It would appear that BCA is the driving factor and not hydrogen sulfide. Both BCA bars are low and removal raises the level. In fact, hydrogen sulfide suppresses relative to LPS but minimally. This is anomalous, but not effectively explained or interpreted.
Line 263 - ‘closely related’ implies a stronger relationship than is revealed by the data. Revise the statement.
Line 273-274 - only IL-8 and IL-1B are reported. Without further examination of other factors it is possible that hydrogen sulfide uses other pathways as well and this possibility should be accounted for in this statement.
Reviewer 4 Report
The authors of the manuscript carried out a study that integrates into the field of interest, and the results are interesting. I make some recommendations:
ü It is necessary to complete the bibliographic data from the specialized literature
ü The conclusions chapter to be completed starting from the results obtained to include a more complete summary information.
Round 2
Reviewer 1 Report
The paper has been considerably improved. Se below for minor changes:
Page 2 lines 80-83: “H2S stimulates leukocyte activation and
trafficking, to promote pulmonary inflammatory response. In the model of septic shock
caused by cecal ligation and perforation [12]. These evidences indicated that H2S may
serve as a dual-role modulator in inflamma-tory process” Remove the dot after “response”; change “inflamma-tory” as “inflammatory”.
Page 3 lines 139-142: “Consistent with the change of mRNA level, protein expression
of IL-8 also showed increased gradually as the increase of LPS concentration in an LPS-
dose dependent manner. while protein expression of IL-1β showed increased gradually
as the increase of LPS concentration” change as “Consistent with the change of mRNA level, protein expression of IL-8 and IL-1β also showed to increase gradually in an LPS- dose dependent manner.”
Page 4 lines 186-187: “However, CBS expression showed with different changes compared to normal MAC-T cells, (Fig. 2 E and G).” remove “with”
Page 8 lines 280-284: “However, for IL-1β, after BCA treatment, its expression was reduced, and NaHS did not reverse this situation, which is inconsistent with the previous data. This may be that the anti-inflammatory effect of the CTH/H2S system does not depend on IL-1β. Whether IL-1β has a direct inhibitory effect, and its specific mechanism of action remains to be further explored” change as “However, IL-1β expression was reduced after BCA treatment, and NaHS did not reverse this situation, which is inconsistent with the previous data. An explanation could be that the anti-inflammatory effect of the CTH/H2S system does not depend on IL-1β. Whether IL-1β has a direct inhibitory effect, and its specific mechanism of action remains to be further explored”
